# An investigation of the performance of parametric functional forms for the Lorenz curve

**Thitithep Sitthiyot** [1]*, **Kanyarat Holasut** [2]

**1** Department of Banking and Finance, Faculty of Commerce and Accountancy, Chulalongkorn University, Bangkok, Thailand, **2** Department of Chemical Engineering, Faculty of Engineering, Khon Kaen University, Khon Kaen, Thailand

* thitithep@cbs.chula.ac.th

**Data Availability Statement:** All data analyzed during this study are included in this article and can be accessed from the Office of the National Economic and Social Development Council (NESDC), Thailand, the United Nations University

## Abstract

Given that an excellent performance of any parametric functional form for the Lorenz curve that is based on a single country case study and a limited range of distribution must be treated with great caution, this study investigates the performance of a single-parameter functional form proposed by Paul and Shankar (2020) who use income data of Australia to show that their functional form is superior to the other existing widely used functional forms considered in their study. By using both mathematical proof and empirical data of 40 countries around the world, this study demonstrates that Paul and Shankar (2020)'s functional form not only fails to fit the actual observations well but also is generally outperformed by the other popular functional forms considered in their study. Moreover, to overcome the limitation of the performance of a single-parameter functional form on the criterion of the estimated Gini index, this study employs a functional form that has more than one parameter in order to show that, by and large, it performs better than all popular single-parameter functional forms considered in Paul and Shankar (2020)'s study. Thus, before applying any functional form to estimate the Lorenz curve, policymakers should check if it could describe the shape of income distributions of different countries through the changes in parameter values and yield the values of the estimated Gini index that are close to their observed data. Using a functional form that does not fit the actual observations could adversely affect inequality measures and income distribution policies.

## Introduction

The Lorenz curve was originally developed by an American economist named Max O. Lorenz in 1905 as a method for measuring wealth concentration [1]. It represents a graphical relationship between the cumulative normalized rank of population from the poorest to the richest and the cumulative normalized wealth held by these population from the poorest to the richest. For more than a century, the Lorenz curve has been widely used for illustrating the distribution of income and for examining inequality in such distribution [2]. It has also performed an

World Institute for Development Economics Research (UNU-WIDER), and the World Bank all of which are listed in References The links to the data sources are as follows: 1. https://www.nesdc.go.th/main.php?filename=social 2. https://www4.wider.unu.edu/ 3. https://data.worldbank.org/.

**Funding:** The authors received no specific funding for this work.

**Competing interests:** Since the findings reported in our original research paper contradict the results previously published in Paul and Shankar (2020), we would like to confirm that we completely agree to policy of PLOS ONE with regard to the Manuscripts Disputing Published Work as stated in PLOS ONE submission guidelines. This does not alter our adherence to PLOS ONE policies on sharing data and materials.

important role in gauging and comparing inequality in income distribution [3] as typically measured by the Gini index which can be computed from the Lorenz curve as shown in Fig 1.

The Gini index can be calculated as the ratio of the area between the perfect equality line and the Lorenz curve (A) divided by the total area under the perfect equality line (A + B) [4]. The Gini index takes the value between 0 and 1. The closer the index is to 0 (where the area A is small), the more equal the distribution of income. The closer the index is to 1 (where the area A is large), the more unequal the distribution of income.

According to Sitthiyot and Holasut [2], the Lorenz curve can be estimated 1) by using interpolation techniques, 2) by assuming a specified functional form for income distribution and deriving the respective Lorenz curve, and 3) by specifying a parametric functional form for the Lorenz curve. Given the interpolation techniques underestimate inequality unless the individual data are available and no existing statistical distribution has proved to be adequate for representing the entire income distribution, numerous studies have proposed a variety of parametric functional forms in order to directly estimate the Lorenz curve [3, 5–31].

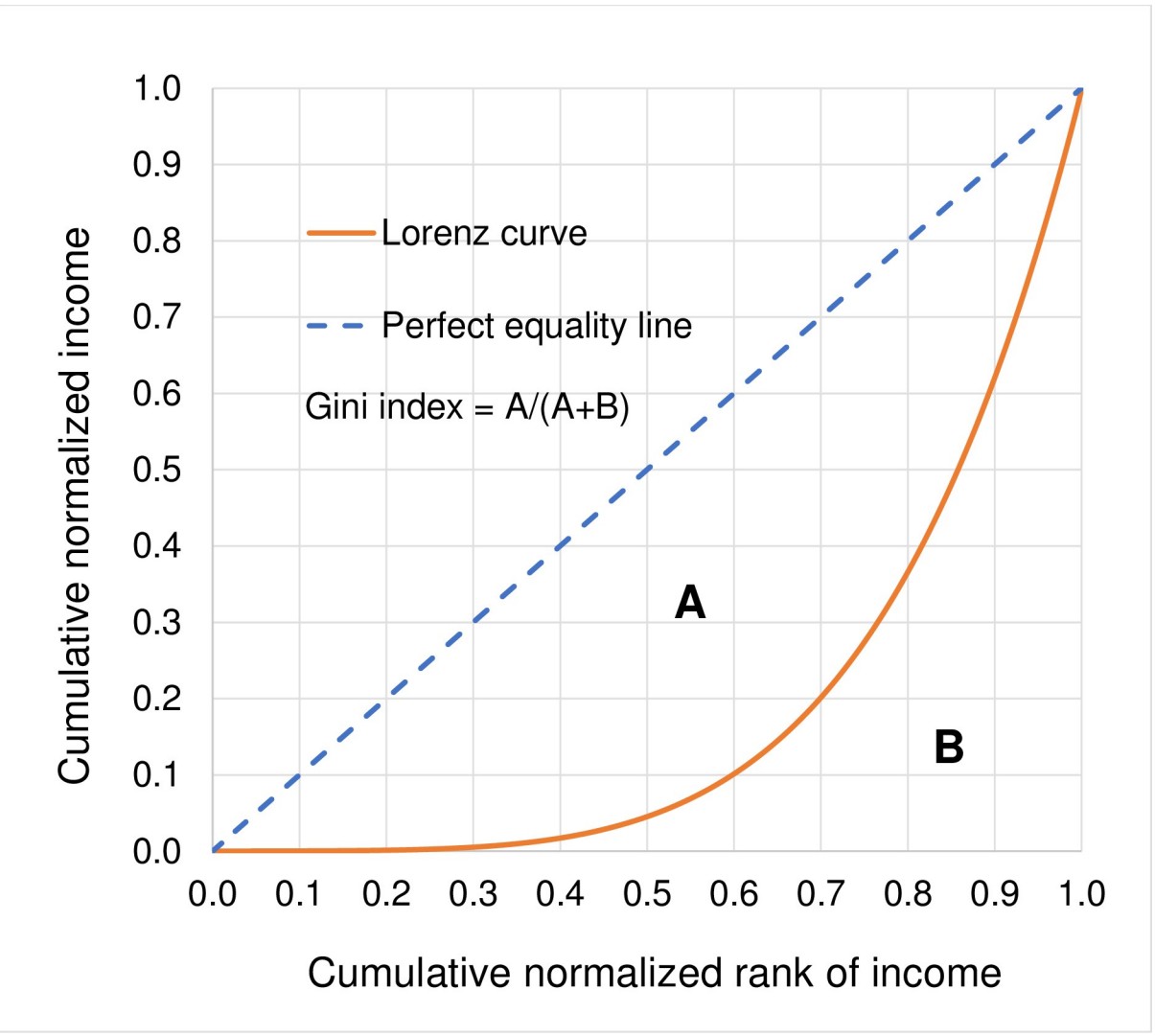

**Fig 1. The Lorenz curve and the Gini index.**

While a number of studies propose a variety of parametric functional forms for estimating the Lorenz curve, the purpose of this study is to investigate an alternative single-parameter functional form for the Lorenz curve proposed by Paul and Shankar [3]. By utilizing the data on the Australian decile income shares between 2001 and 2010 that are constructed using the individual income data from the first 10 waves of Household, Income, and Labour Dynamics Australia Survey, Paul and Shankar [3] show that, on the basis of mean-squared error (MSE) and information inaccuracy measure (IIM), their functional form outperforms the other existing widely used functional forms, namely, Kakwani and Podder [5], Aggarwal [9], Chotikapanich [16], and a functional form implied by Pareto distribution. Moreover, the Gini index in all 10 years, calculated using the functional form that Paul and Shankar [3] propose, rank second closely behind those estimated using Aggarwal [9]'s functional form.

According to Jordá et al. [32], an excellent performance of any parametric functional form for the Lorenz curve that is relied on a single country case study and a limited range of distribution should be treated with extreme caution. Jordá et al. [32] also note that previous studies by [33, 34] also point out the similar caveat. Given the superiority of Paul and Shankar [3]'s functional form over the other existing widely used functional forms, it is therefore relevant and worthwhile to conduct an examination to find out if we use grouped income data of other countries, the performance of Paul and Shankar [3]'s functional form is *still* superior to the other existing widely used functional forms considered in their study. The findings from this investigation should also serve as a check-and-balance not only for economics but also for other scientific disciplines that use the Lorenz curve to analyze size distributions of non-negative quantities and inequalities. This is because a good functional form for the Lorenz curve must describe the shape of income distributions of different countries, regions, socioeconomic groups, and in different time periods through the changes in parameter values [35]. In addition, it should provide a good fit of the entire range of income distribution since all observations are relevant for an accurate measurement of income inequality, supporting social and income policies, as well as determining taxation structure [35]. Provided that various studies on the relationship between inequality measures and financial and/or socioeconomic variables, for example [36–39], rely on the accuracy of inequality measures that could possibly be derived from a parametric functional form for the Lorenz curve, if the choice of parametric functional form is not a valid candidate for representing the income distribution, the estimates on the income shares and inequality measures might be severely affected by misspecification bias [32].

Our findings indicate that, despite its parsimony, the single-parameter functional form for the Lorenz curve proposed by Paul and Shankar [3] has two serious limitations. First, it fails to fit the actual observations well when the inequality in income distribution as measured by the Gini index is lower than the critical threshold which is found to be 0.4180. We verify the first limitation by demonstrating both mathematically and empirically that when a country's Gini index is lower than 0.4180, not only does the functional form proposed by Paul and Shankar [3] not perform well but also all other popular functional forms considered in their study, by and large, perform far better than Paul and Shankar [3]'s specified functional form on the basis of the coefficient of determination ($R^2$), MSE, IIM, and the estimated Gini index. The numerical example and the income data of 20 Organization for Economic Co-operation and Development (OECD) countries are employed in order to illustrate the first limitation. Second, on the basis of $R^2$, MSE, IIM, and the estimated Gini index, the performance of Paul and Shankar [3]'s functional specification, compared to the other existing widely used functional forms considered in their study, is at best mixed when the upper part of a country's income distribution has a long-tailed property which is defined as the income share of the top 20% being greater than 50% of total income share. The income data of the other 20 different

countries around the world whose upper part of income distribution has a long-tailed property are employed in order to illustrate the second limitation.

While Paul and Shankar [3] show that the values of the Gini index estimated using the functional form that they propose rank second closely behind those estimated using Aggarwal [9]'s functional form, this study would like to note that, to evaluate the performance of a functional form for the Lorenz curve on the basis of the estimated Gini index, a functional form that contains more than one parameter is required since the curvature of the estimated Lorenz curve has to be adjustable so that it would fit the actual data as much as possible while keeping the value of the estimated Gini index the same. According to Dagum [35], this cannot be done by using a single-parameter functional form because the estimated Gini index would be a monotonic function of it. In order to illustrate that a functional form that has more than one parameter, by and large, gives the values of the estimated Gini index closer to the actual observations than single-parameter functional forms, a two-parameter functional form proposed by Sitthiyot and Holasut [2] is employed in order to estimate the Lorenz curve and calculate the value of the estimated Gini index using the same set of grouped income data of 40 countries.

## Materials and methods

To examine the performance of Paul and Shankar [3]'s functional form, let $p$ denote the cumulative normalized rank of income and $L(p,\gamma)$ denote the cumulative normalized income for a given value of parameter $\gamma$. The functional form for the Lorenz curve proposed by Paul and Shankar [3] is as follows:

$$L(p,\gamma) = p * \left[\frac{e^{-\gamma(1-e^p)} - 1}{e^{-\gamma(1-e)} - 1}\right], \gamma > 0 \tag{1}$$

Given that the value of parameter $\gamma$ must be greater than 0, the first limitation of this functional form could be demonstrated mathematically by setting the value of parameter $\gamma$ to be 0 and then using Eq (1) to calculate the values of $L(p,\gamma)$ for different values of $p$ which are 0.0, 0.1, 0.2, 0.3, 0.4, 0.5, 0.6, 0.7, 0.8, 0.9, and 1.0. All estimated values of $L(p,\gamma)$ would be used to calculate the area under the Lorenz curve using the numerical integration since the above specified functional form does not have a closed-form expression for the Gini index. The next step is to compute the value of the Gini index that corresponds to the value of parameter $\gamma = 0$. The value of the Gini index, given $\gamma = 0$, would be the critical threshold where the specified functional form would not be able to fit the actual observations well below this critical threshold since the value of parameter $\gamma$ cannot be 0 or negative.

To illustrate the first limitation of Paul and Shankar [3]'s functional form empirically as well as to evaluate its performance compared to those of the other popular functional specifications considered in their study, namely, Kakwani and Podder [5], Aggarwal [9], Chotikapanich [16], and a functional form implied by Pareto distribution, this study uses the data on the observed Gini index and the decile income shares of 20 OECD countries from the United Nations University-World Income Inequality Database (UNU-WIID) [40]. The reason that we choose the OECD countries is because the values of the observed Gini index of these countries are lower than the critical threshold. We would like to note that, based on the mathematical proof as described above, this critical threshold should apply to not only income distribution of any country and in any time period but also income distribution generated by using any simulation method as long as the value of the Gini index is less than 0.4180, not just to the income distributions of 20 OECD countries employed in this study.

For the second limitation of Paul and Shankar [3]'s functional form where it does not perform well compared to the other popular functional forms when a country's income

distribution has a long-tailed property, the data on the observed Gini index and the income shares by decile of the other 20 different countries around the world from the UNU-WIID [40], the Office of the National Economic and Social Development Council (NESDC), Thailand [41], and the World Bank [42] are used to illustrate this limitation. These countries are chosen because the upper part of income distribution is long-tailed where the top 20% of income earners receive more than 50% of the total income as defined in Introduction.

In this study, the performance of functional form proposed by Paul and Shankar [3] is compared to the other existing widely used functional specifications considered in their study which are Kakwani and Podder [5], Aggarwal [9], Chotikapanich [16], and a functional form implied by Pareto distribution. These functional forms are expressed as follows:

Kakwani and Podder [5]:

$$L(p, \delta) = p e^{-\delta(1-p)}, \delta > 0 \tag{2}$$

Aggarwal [9]:

$$L(p, \theta) = \frac{(1-\theta)^2 p}{(1+\theta)^2 - 4\theta p}, 0 < \theta < 1 \tag{3}$$

Chotikapanich [16]:

$$L(p, k) = \frac{e^{kp} - 1}{e^k - 1}, k > 0 \tag{4}$$

Functional form implied by Pareto distribution:

$$L(p, \alpha) = 1 - (1-p)^{\frac{1}{\alpha}}, \alpha > 1 \tag{5}$$

Paul and Shankar [3, p. 1396] note that their alternative functional form for the Lorenz curve, Kakwani and Podder [5]'s functional form, and a functional form implied by Pareto distribution do not have a closed-form expression for the Gini index, hence the estimated Gini index has to be calculated by using the numerical integration. Provided that a good functional form for the Lorenz curve should have a closed-form expression for the Gini index [35], this study finds out that only does Paul and Shankar [3]'s functional form not have an explicit mathematical solution for the Gini index. For the other existing widely used functional forms considered in their study, the Gini index can be directly computed since these functional forms have closed-form expressions for the Gini index which are as follows:

Kakwani and Podder [5]:

$$\text{Gini index} = 1 - \frac{2(\delta - 1)}{\delta^2} - \frac{2e^{-\delta}}{\delta^2} \tag{6}$$

Aggarwal [9]:

$$\text{Gini index} = \frac{(1+\theta)^2}{2\theta} \left[ \frac{(1-\theta)^2}{4\theta} ln \left( \frac{1-\theta}{1+\theta} \right)^2 + 1 \right] - 1 \tag{7}$$

Chotikapanich [16]:

$$\text{Gini index} = \frac{(k-2)e^k + (k+2)}{k(e^k - 1)} \tag{8}$$

**Table 1. The descriptive statistics of the observed Gini index and the decile income shares of 20 OECD countries whose values of the observed Gini index are below the critical threshold of 0.4180.**

| Indicator | Observed Gini index | Decile income shares | | | | | | | | | |
|---|---|---|---|---|---|---|---|---|---|---|---|
| | | D1 | D2 | D3 | D4 | D5 | D6 | D7 | D8 | D9 | D10 |
| Mean | 0.2810 | 0.0326 | 0.0536 | 0.0646 | 0.0744 | 0.0843 | 0.0948 | 0.1065 | 0.1212 | 0.1434 | 0.2247 |
| Median | 0.2805 | 0.0320 | 0.0540 | 0.0645 | 0.0745 | 0.0840 | 0.0950 | 0.1065 | 0.1210 | 0.1420 | 0.2255 |
| Mode | 0.3090 | 0.0320 | 0.0580 | 0.0640 | 0.0760 | 0.0850 | 0.0950 | 0.1080 | 0.1210 | 0.1420 | 0.2240 |
| Minimum | 0.2320 | 0.0200 | 0.0460 | 0.0570 | 0.0680 | 0.0790 | 0.0910 | 0.1020 | 0.1160 | 0.1360 | 0.1910 |
| Maximum | 0.3270 | 0.6510 | 0.0620 | 0.0740 | 0.0820 | 0.0930 | 0.1000 | 0.1110 | 0.1300 | 0.1590 | 0.2450 |
| Standard deviation | 0.0268 | 0.0053 | 0.0046 | 0.0044 | 0.0037 | 0.0032 | 0.0024 | 0.0024 | 0.0033 | 0.0061 | 0.0152 |
| Number of countries | 20 | 20 | 20 | 20 | 20 | 20 | 20 | 20 | 20 | 20 | 20 |

Functional form implied by Pareto distribution:

$$\text{Gini index} = \frac{\alpha - 1}{\alpha + 1} \tag{9}$$

This study uses the curve fitting technique based on minimizing error sum of squares to estimate the Lorenz curve. The statistical measures of goodness-of-fit, namely, MSE and IIM are employed for assessing the performance of all functional forms, both of which are the same criteria as used in Paul and Shankar [3]'s study. In addition, this study uses $R^2$ and the estimated Gini index to evaluate the performance of all functional forms. The closer the value of $R^2$ is to 1 and the closer the value of MSE is to zero, the better the estimated functional form. For the IIM criterion, the estimated functional specification that has a smaller absolute value of IIM is better than those with larger absolute values of IIM. In this study, the Microsoft Excel Data Analysis program and the Microsoft Excel Solver program are used for calculating the descriptive statistics on the Gini index and the income shares by decile as well as estimating the parameters and calculating the estimated Gini index. As suggested by Dagum [35], from a perspective of computational cost and the acceptance of the specified functional form in practice, a simple method of parameter estimation is always an advantage. Table 1 reports the descriptive statistics of the observed Gini index and the decile income shares of 20 OECD countries whereas those of the other 20 countries are reported in Table 2.

## Results

### The mathematical proof of the critical threshold for the Gini index

This study first demonstrates the mathematical proof of the first limitation of Paul and Shankar [3]'s functional form. According to their specified functional form with the values of $p$

**Table 2. The descriptive statistics of the observed Gini index and the decile income shares of the other 20 countries whose upper part of the income distribution has a long-tailed property defined as the income share of the top 20% being greater than 50% of total income share.**

| Indicator | Income share of the top 20% | Observed Gini index | Decile income shares | | | | | | | | | |
|---|---|---|---|---|---|---|---|---|---|---|---|---|
| | | | D1 | D2 | D3 | D4 | D5 | D6 | D7 | D8 | D9 | D10 |
| Mean | 0.5576 | 0.5065 | 0.0140 | 0.0263 | 0.0363 | 0.0455 | 0.0561 | 0.0688 | 0.0852 | 0.1100 | 0.1545 | 0.4032 |
| Median | 0.5541 | 0.5070 | 0.0126 | 0.0257 | 0.0348 | 0.0452 | 0.0561 | 0.0703 | 0.0858 | 0.1110 | 0.1572 | 0.3989 |
| Mode | N/A | 0.5400 | 0.0100 | 0.0260 | 0.0342 | N/A | 0.0600 | 0.0680 | 0.0870 | 0.1150 | 0.1510 | N/A |
| Minimum | 0.5112 | 0.4470 | 0.0090 | 0.0179 | 0.0250 | 0.0331 | 0.0428 | 0.0556 | 0.0742 | 0.0979 | 0.1331 | 0.3529 |
| Maximum | 0.6374 | 0.5907 | 0.0264 | 0.0375 | 0.0452 | 0.0537 | 0.0646 | 0.0769 | 0.0933 | 0.1180 | 0.1690 | 0.4725 |
| Standard deviation | 0.0322 | 0.0359 | 0.0043 | 0.0043 | 0.0047 | 0.0049 | 0.0050 | 0.0051 | 0.0053 | 0.0060 | 0.0108 | 0.0328 |
| Number of countries | 20 | 20 | 20 | 20 | 20 | 20 | 20 | 20 | 20 | 20 | 20 | 20 |

**Table 3. The values of the Gini index calculated using different values of parameter $\gamma$ based on Paul and Shankar [3]'s functional form.**

| Parameter $\gamma$ | Gini index |
|---|---|
| -0.000001 | 0.4179801 |
| **0** | **0.4179803** |
| 0.000001 | 0.4179804 |
| 0.00001 | 0.4179818 |
| 0.0001 | 0.4179955 |
| 0.001 | 0.4181326 |
| 0.01 | 0.4195044 |
| 0.1 | 0.4332468 |
| 0.5 | 0.4941095 |
| 1.0 | 0.5665420 |
| 1.25 | 0.6000009 |
| 1.5 | 0.6311545 |
| 2.0 | 0.6859776 |

being equal to 0.0, 0.1, 0.2, 0.3, 0.4, 0.5, 0.6, 0.7, 0.8, 0.9, and 1.0, Table 3 reports different values of the Gini index (rounding up to 7 decimal places) for different values of parameter $\gamma$ ranging between −0.000001 and 2.0.

The numerical results in Table 3 indicate that when the value of parameter $\gamma$ is equal to 0 (the average value of parameter $\gamma$ when $\gamma$ = −0.000001 and 0.000001), the critical threshold for the Gini index is equal to 0.4179803 (the average value of the Gini index when the Gini index = 0.4179801 and 0.4179804). Let the value of the critical threshold for the Gini index = 0.4180 for the purpose of demonstration. This implies that when the level of inequality in income distribution as measured by the observed Gini index is less than the critical threshold, using Paul and Shankar [3]'s functional form for approximating the Lorenz curve and calculating the Gini index would result in the value of the estimated Gini index being equal to 0.4180. Provided that this is the mathematical proof, it is thus always true not only for income distribution of any country and in any time period but also income distribution generated by using any simulation technique so long as the value of the Gini index is less than 0.4180.

### The performance comparison between Paul and Shankar [3]'s functional form and the other popular functional forms when the Gini index is below the critical threshold

In order to show the first limitation of Paul and Shankar [3]'s functional form empirically, this study employs the data on decile income shares of 20 OECD countries from the UNU-WIID [40] and estimate the Lorenz curve using their functional form and the other existing widely used functional specifications considered in their study, namely, Kakwani and Podder [5], Aggarwal [9], Chotikapanich [16], and a functional form implied by Pareto distribution. The estimated values of parameters for each functional form are reported in Table A1 in S1 Appendix. We then compare the performance of Paul and Shankar [3]'s functional form to those of the other existing widely used functional forms on the criteria of $R^2$, MSE, IIM, and the estimated Gini index. Tables 4–7 reports our results.

The overall results in Tables 4–6 suggest that, on the criteria of $R^2$, MSE, and IIM, all other popular functional forms perform far better than the functional form proposed by Paul and Shankar [3]. With regard to the value of the estimated Gini index, all results in Table 7 clearly confirm the first limitation of Paul and Shankar [3]'s functional form in that when the

**Table 4. The income data of 20 OECD countries are used to demonstrate the performance of Paul and Shankar [3]'s functional form and those of the other popular functional forms based on $R^2$.**

| Country | Year | Observed Gini index | Coefficient of determination ($R^2$) | | | | |
|---|---|---|---|---|---|---|---|
| | | | PS | KP | A | C | Pareto |
| Slovakia | 2017 | 0.2320 | 0.8760 | 0.9989 | 0.9986 | 0.9988 | 0.9918 |
| Slovenia | 2017 | 0.2370 | 0.8827 | 0.9990 | 0.9993 | 0.9992 | 0.9931 |
| Czech Republic | 2017 | 0.2450 | 0.8851 | 0.9974 | 0.9989 | 0.9978 | 0.9958 |
| Finland | 2017 | 0.2530 | 0.8958 | 0.9975 | 0.9990 | 0.9980 | 0.9951 |
| Belgium | 2017 | 0.2600 | 0.9094 | 0.9989 | 0.9992 | 0.9991 | 0.9915 |
| Norway | 2017 | 0.2610 | 0.9039 | 0.9975 | 0.9984 | 0.9978 | 0.9934 |
| Netherlands | 2017 | 0.2710 | 0.9151 | 0.9978 | 0.9989 | 0.9982 | 0.9933 |
| Denmark | 2017 | 0.2760 | 0.9151 | 0.9967 | 0.9983 | 0.9972 | 0.9940 |
| Austria | 2017 | 0.2790 | 0.9244 | 0.9978 | 0.9987 | 0.9981 | 0.9921 |
| Sweden | 2017 | 0.2800 | 0.9245 | 0.9979 | 0.9987 | 0.9982 | 0.9918 |
| Hungary | 2017 | 0.2810 | 0.9270 | 0.9979 | 0.9989 | 0.9983 | 0.9920 |
| Germany | 2017 | 0.2910 | 0.9346 | 0.9974 | 0.9989 | 0.9979 | 0.9921 |
| Poland | 2017 | 0.2920 | 0.9349 | 0.9971 | 0.9987 | 0.9977 | 0.9924 |
| France | 2017 | 0.2930 | 0.9295 | 0.9950 | 0.9978 | 0.9958 | 0.9948 |
| Switzerland | 2017 | 0.3010 | 0.9394 | 0.9959 | 0.9983 | 0.9966 | 0.9934 |
| Ireland | 2017 | 0.3060 | 0.9464 | 0.9966 | 0.9990 | 0.9974 | 0.9926 |
| Canada | 2017 | 0.3090 | 0.9532 | 0.9983 | 0.9988 | 0.9986 | 0.9885 |
| Luxembourg | 2017 | 0.3090 | 0.9524 | 0.9978 | 0.9991 | 0.9983 | 0.9899 |
| Estonia | 2017 | 0.3160 | 0.9618 | 0.9991 | 0.9989 | 0.9993 | 0.9857 |
| Italy | 2017 | 0.3270 | 0.9639 | 0.9978 | 0.9982 | 0.9981 | 0.9869 |

PS, Paul and Shankar [3]; KP, Kakwani and Podder [5]; A, Aggarwal [9]; C, Chotikapanich [16]; Pareto, Functional form implied by Pareto distribution.

The bold numbers (if any) indicate that Paul and Shankar [3]'s functional form is superior to the other existing widely used functional forms.

observed Gini index is lower than the critical threshold, using Paul and Shankar [3]'s functional form to estimate the Lorenz curve and compute the Gini index would result in the estimated Gini index being equal to 0.4180. Provided that the parameter $\gamma$ cannot take value below 0, this study sets the value of estimated parameter $\gamma$ to be equal to 0.0000001 which would result in the estimated value of the Gini index to be 0.4180 for all 20 OECD countries. The results in Table 7 also show that the values of the estimated Gini index calculated using the other popular functional forms, namely, Kakwani and Podder [5], Aggarwal [9], Chotikapanich [16], and a functional form implied by Pareto distribution are much closer to the actual observations than those calculated using Paul and Shankar [3]'s functional specification. The performance comparison between Paul and Shankar [3]'s functional forms and the other existing widely used functional forms considered in their study when the observed Gini index is less than the critical threshold of 0.4180 is also illustrated in Fig 2.

## The performance comparison between Paul and Shankar [3]'s functional form and the other popular functional forms when the upper part of income distribution has a long-tailed property

To demonstrate the second limitation of Paul and Shankar [3]'s functional specification, this study employs the data on income shares by decile of the other 20 different countries around the world from the UNU-WIID [40], the NESDC [41], and the World Bank [42] to estimate the Lorenz curve. Table A2 in S1 Appendix reports the estimated values of parameters for each

**Table 5. The income data of 20 OECD countries are used to demonstrate the performance of Paul and Shankar [3]'s functional form and those of the other popular functional forms based on MSE.**

| Country | Year | Observed Gini index | Mean-squared error (MSE) | | | | |
|---|---|---|---|---|---|---|---|
| | | | **PS** | **KP** | **A** | **C** | **Pareto** |
| Slovakia | 2017 | 0.2320 | 0.0012 | 0.0001 | 0.0001 | 0.0001 | 0.0003 |
| Slovenia | 2017 | 0.2370 | 0.0011 | 0.0001 | 0.0001 | 0.0001 | 0.0003 |
| Czech Republic | 2017 | 0.2450 | 0.0011 | 0.0002 | 0.0001 | 0.0002 | 0.0002 |
| Finland | 2017 | 0.2530 | 0.0010 | 0.0002 | 0.0001 | 0.0002 | 0.0002 |
| Belgium | 2017 | 0.2600 | 0.0009 | 0.0001 | 0.0001 | 0.0001 | 0.0004 |
| Norway | 2017 | 0.2610 | 0.0010 | 0.0002 | 0.0002 | 0.0002 | 0.0003 |
| Netherlands | 2017 | 0.2710 | 0.0008 | 0.0002 | 0.0001 | 0.0002 | 0.0003 |
| Denmark | 2017 | 0.2760 | 0.0009 | 0.0003 | 0.0002 | 0.0003 | 0.0002 |
| Austria | 2017 | 0.2790 | 0.0008 | 0.0002 | 0.0001 | 0.0002 | 0.0003 |
| Sweden | 2017 | 0.2800 | 0.0008 | 0.0002 | 0.0001 | 0.0002 | 0.0003 |
| Hungary | 2017 | 0.2810 | 0.0008 | 0.0002 | 0.0001 | 0.0002 | 0.0004 |
| Germany | 2017 | 0.2910 | 0.0007 | 0.0002 | 0.0001 | 0.0002 | 0.0003 |
| Poland | 2017 | 0.2920 | 0.0007 | 0.0002 | 0.0001 | 0.0002 | 0.0003 |
| France | 2017 | 0.2930 | 0.0009 | 0.0004 | 0.0002 | 0.0004 | 0.0002 |
| Switzerland | 2017 | 0.3010 | 0.0008 | 0.0004 | 0.0002 | 0.0003 | 0.0003 |
| Ireland | 2017 | 0.3060 | 0.0006 | 0.0003 | 0.0001 | 0.0002 | 0.0004 |
| Canada | 2017 | 0.3090 | 0.0005 | 0.0002 | 0.0001 | 0.0002 | 0.0005 |
| Luxembourg | 2017 | 0.3090 | 0.0006 | 0.0002 | 0.0001 | 0.0002 | 0.0005 |
| Estonia | 2017 | 0.3160 | 0.0004 | 0.0001 | 0.0001 | 0.0001 | **0.0007** |
| Italy | 2017 | 0.3270 | 0.0005 | 0.0002 | 0.0001 | 0.0002 | **0.0006** |

PS, Paul and Shankar [3]; KP, Kakwani and Podder [5]; A, Aggarwal [9]; C, Chotikapanich [16]; Pareto, Functional form implied by Pareto distribution.

The bold numbers (if any) indicate that Paul and Shankar [3]'s functional form is superior to the other existing widely used functional forms.

functional form. We then evaluate the performance of Paul and Shankar [3]'s functional form as well as those of the other existing widely used functional forms considered in their study based on $R^2$, MSE, IIM, and the estimated Gini index. Our results are reported in Tables 8–11.

The overall results in Tables 8–11 show that Paul and Shankar [3]'s functional form is, by and large, outperformed by the other existing widely used functional forms, namely, Kakwani and Podder [5], Aggarwal [9], Chotikapanich [16], and a functional form implied by Pareto distribution. The results in Table 8 show that, on the basis of $R^2$, there are *only* 2 out of 100 cases where Paul and Shankar [3]'s functional form performs better than the other existing widely used functional forms. Recall that Paul and Shankar [3] show that, on the criteria of MSE and IIM, their functional form fits the income data of Australia better than the other existing widely used functional forms considered in their study. However, using the income data of the other 20 countries and the same performance assessment criteria as those used in Paul and Shankar [3]'s study, the results in Tables 9 and 10 indicate that their functional form is superior to the other existing widely used functional forms in *only* 9 out of 100 cases based on MSE and *only* 2 out of 100 cases based on IIM, respectively. Lastly, on the criterion of the estimated Gini index as measured by the absolute value of the difference between the estimated Gini index and the observed Gini index, the results in Table 11 show that the functional form proposed by Paul and Shankar [3] outperforms the other popular functional specifications in 22 out of 100 cases. Fig 3 illustrates the performance comparison between Paul and Shankar [3]'s functional form and the other existing widely used functional forms considered in their study when the income data has a long-tailed property.

**Table 6. The income data of 20 OECD countries are used to demonstrate the performance of Paul and Shankar [3]'s functional form and those of the other popular functional forms based on IIM.**

| Country | Year | Observed Gini index | Information inaccuracy measure (IIM) | | | | |
|---|---|---|---|---|---|---|---|
| | | | PS | KP | A | C | Pareto |
| Slovakia | 2017 | 0.2320 | 0.1014 | 0.0077 | 0.0087 | 0.0079 | 0.0201 |
| Slovenia | 2017 | 0.2370 | 0.1045 | 0.0046 | 0.0044 | 0.0045 | 0.0148 |
| Czech Republic | 2017 | 0.2450 | 0.1078 | 0.0050 | 0.0025 | 0.0043 | 0.0062 |
| Finland | 2017 | 0.2530 | 0.1016 | 0.0056 | 0.0032 | 0.0049 | 0.0083 |
| Belgium | 2017 | 0.2600 | 0.0850 | 0.0041 | 0.0040 | 0.0040 | 0.0168 |
| Norway | 2017 | 0.2610 | 0.0885 | 0.0095 | 0.0084 | 0.0092 | 0.0150 |
| Netherlands | 2017 | 0.2710 | 0.0845 | 0.0070 | 0.0052 | 0.0065 | 0.0139 |
| Denmark | 2017 | 0.2760 | 0.0854 | 0.0106 | 0.0082 | 0.0100 | 0.0131 |
| Austria | 2017 | 0.2790 | 0.0762 | 0.0089 | 0.0077 | 0.0086 | 0.0180 |
| Sweden | 2017 | 0.2800 | 0.0753 | 0.0076 | 0.0065 | 0.0073 | 0.0168 |
| Hungary | 2017 | 0.2810 | 0.0751 | 0.0084 | 0.0070 | 0.0080 | 0.0182 |
| Germany | 2017 | 0.2910 | 0.0710 | 0.0077 | 0.0056 | 0.0071 | 0.0162 |
| Poland | 2017 | 0.2920 | 0.0704 | 0.0085 | 0.0061 | 0.0078 | 0.0164 |
| France | 2017 | 0.2930 | 0.0825 | 0.0142 | 0.0095 | 0.0130 | 0.0119 |
| Switzerland | 2017 | 0.3010 | 0.0707 | 0.0102 | 0.0062 | 0.0091 | 0.0127 |
| Ireland | 2017 | 0.3060 | 0.0693 | 0.0093 | 0.0052 | 0.0081 | 0.0158 |
| Canada | 2017 | 0.3090 | 0.0518 | 0.0053 | 0.0049 | 0.0050 | 0.0229 |
| Luxembourg | 2017 | 0.3090 | 0.0546 | 0.0046 | 0.0029 | 0.0040 | 0.0188 |
| Estonia | 2017 | 0.3160 | 0.0429 | 0.0029 | 0.0040 | 0.0029 | 0.0297 |
| Italy | 2017 | 0.3270 | 0.0423 | 0.0099 | 0.0102 | 0.0098 | 0.0296 |

PS, Paul and Shankar [3]; KP, Kakwani and Podder [5]; A, Aggarwal [9]; C, Chotikapanich [16]; Pareto, Functional form implied by Pareto distribution.

The bold numbers (if any) indicate that Paul and Shankar [3]'s functional form is superior to the other existing widely used functional forms.

Although one of the aims to develop a functional form for the Lorenz curve is to compute inequality measures such as the Gini index that would be close to the actual observation [3], we would like to note that the shape of income distribution is also important and should be taken into consideration when evaluating the performance of different functional forms. This is because it is possible that different functional forms could have the estimated Gini index that is identical to its observed value but the shapes of distributions are different. By using the income data of Thailand, this point could be demonstrated by finding the value of the estimated parameter of functional form proposed by Paul and Shankar [3] and those of the other popular functional forms that result in the same value of the estimated Gini index. The results are reported in Table 12.

The results in Table 12 show that the estimated parameters of all functional forms are different but all of them result in the same estimated Gini index which is 0.4528. Fig 4 illustrates the actual Lorenz plot and the estimated Lorenz curves using Paul and Shankar [3]'s functional form and the other existing widely used functional forms all of which share the same estimated Gini index.

Given that there are infinite number of the Lorenz curves that could yield the same value of the Gini index, a good functional form should be able to describe the shape of income distributions through the changes in the value of parameters and the fact that it fits the actual data would be the main reason for its choice [35]. Therefore, the shape of income distribution is relatively more important than how close the estimated Gini index is to its actual observation and the priority should be given to the values of goodness-of-fit statistics when assessing the

**Table 7. The income data of 20 OECD countries are used to demonstrate the performance of Paul and Shankar [3]'s functional form and those of the other popular functional forms based on the estimated Gini index.**

| Country | Year | Observed Gini index | Estimated Gini index | | | | |
|---|---|---|---|---|---|---|---|
| | | | **PS** | **KP** | **A** | **C** | **Pareto** |
| Slovakia | 2017 | 0.2320 | 0.4180 | 0.2242 | 0.2227 | 0.2239 | 0.2171 |
| Slovenia | 2017 | 0.2370 | 0.4180 | 0.2315 | 0.2305 | 0.2313 | 0.2250 |
| Czech Republic | 2017 | 0.2450 | 0.4180 | 0.2386 | 0.2389 | 0.2387 | 0.2351 |
| Finland | 2017 | 0.2530 | 0.4180 | 0.2470 | 0.2470 | 0.2470 | 0.2426 |
| Belgium | 2017 | 0.2600 | 0.4180 | 0.2540 | 0.2526 | 0.2538 | 0.2463 |
| Norway | 2017 | 0.2610 | 0.4180 | 0.2520 | 0.2514 | 0.2518 | 0.2467 |
| Netherlands | 2017 | 0.2710 | 0.4180 | 0.2628 | 0.2622 | 0.2627 | 0.2568 |
| Denmark | 2017 | 0.2760 | 0.4180 | 0.2646 | 0.2644 | 0.2646 | 0.2599 |
| Austria | 2017 | 0.2790 | 0.4180 | 0.2711 | 0.2701 | 0.2709 | 0.2644 |
| Sweden | 2017 | 0.2800 | 0.4180 | 0.2709 | 0.2697 | 0.2706 | 0.2639 |
| Hungary | 2017 | 0.2810 | 0.4180 | 0.2739 | 0.2728 | 0.2737 | 0.2669 |
| Germany | 2017 | 0.2910 | 0.4180 | 0.2833 | 0.2824 | 0.2831 | 0.2764 |
| Poland | 2017 | 0.2920 | 0.4180 | 0.2843 | 0.2835 | 0.2842 | 0.2777 |
| France | 2017 | 0.2930 | 0.4180 | 0.2834 | 0.2837 | 0.2834 | 0.2794 |
| Switzerland | 2017 | 0.3010 | 0.4180 | 0.2927 | 0.2924 | 0.2926 | 0.2871 |
| Ireland | 2017 | 0.3060 | 0.4180 | 0.3002 | 0.2995 | 0.3001 | 0.2933 |
| Canada | 2017 | 0.3090 | 0.4180 | 0.3027 | 0.3004 | 0.3022 | 0.2929 |
| Luxembourg | 2017 | 0.3090 | 0.4180 | 0.3037 | 0.3019 | 0.3034 | 0.2948 |
| Estonia | 2017 | 0.3160 | 0.4180 | 0.3120 | 0.3087 | 0.3114 | 0.2999 |
| Italy | 2017 | 0.3270 | 0.4180 | 0.3183 | 0.3156 | 0.3178 | 0.3079 |

PS, Paul and Shankar [3]; KP, Kakwani and Podder [5]; A, Aggarwal [9]; C, Chotikapanich [16]; Pareto, Functional form implied by Pareto distribution.

The bold numbers (if any) indicate that Paul and Shankar [3]'s functional form is superior to the other existing widely used functional forms.

performance of different functional forms. Viewed this way, the performance of Paul and Shankar [3]'s functional form compared to the other existing widely used functional forms considered in their study on the basis of the estimated Gini index, is at best mixed. It does not rank second closely behind that of Aggarwal [9] as for the case when using the income data of Australia reported in their study.

## The performance comparison on the basis of the estimated Gini index between single-parameter functional forms and a functional form that contains more than one parameter

Although the key advantage of any single-parameter functional form is its parsimony, we would like to point out that, in order to evaluate the performance of a functional form for the Lorenz curve on the criteria of the estimated Gini index, besides the issues of the shape of income distribution and the goodness-of-fit statistics as discussed above, a functional form that contains more than one parameter is required since the curvature of the estimated Lorenz curve has to be adjustable so that it would fit the actual observations as much as possible while keeping the value of the estimated Gini index constant. As noted by Dagum [35], this cannot be done by using a single-parameter functional form since the estimated Gini index would be a monotonic function of it.

In order to illustrate that a functional form that contains more than one parameter could generally be used to compute the value of the estimated Gini index that would be closer to the actual observations than single-parameter functional forms, this study employs a two-parameter functional form proposed by Sitthiyot and Holasut [2] to estimate the Lorenz curve using

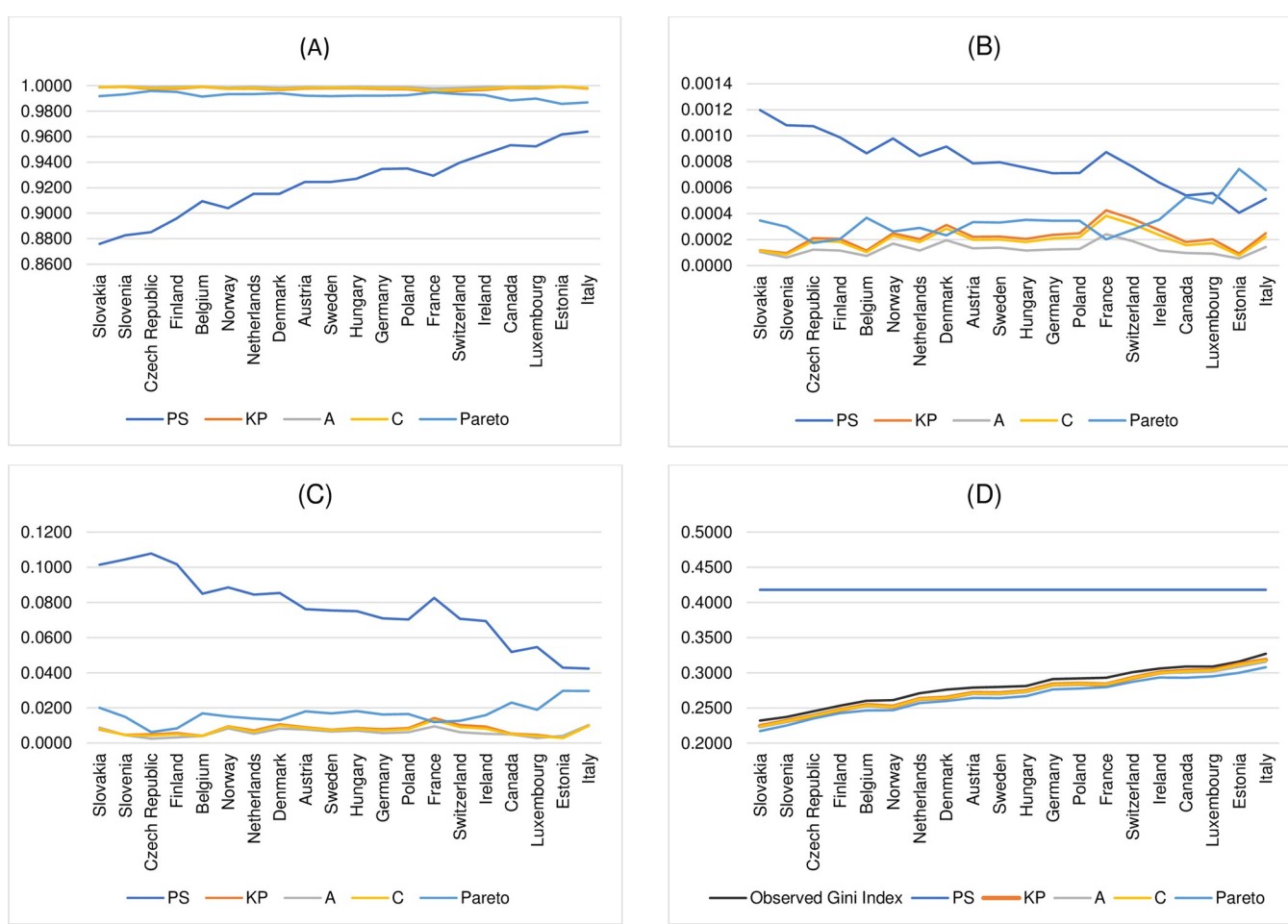

**Fig 2. The performance comparison between Paul and Shankar [3]'s functional form and the other existing widely used functional forms, namely, Kakwani and Podder [5], Aggarwal [9], Chotikapanich [16], and functional form implied by Pareto distribution when the observed Gini index is less than the critical threshold of 0.4180.** (A) R$^2$. (B) MSE. (C) IIM. (D) Estimated Gini index.

the same set of data on decile income shares of 40 countries. The reason that we choose Sitthiyot and Holasut [2]'s functional specification is not only because their specified functional form has several advantages over the functional form proposed by Kakwani [7] which, according to Cheong [43] and Tanak et al. [31], has the best overall performance among all popular functional forms for the Lorenz curves but also has an explicit mathematical solution for the Gini index. By using the same notations for the cumulative normalized rank of income ($p$) and the cumulative normalized income $L(p)$ as denoted in Materials and Methods and also let $a$ and $b$ be parameters, the specified functional form for the Lorenz curve proposed by Sitthiyot and Holasut [2] is as follows:

$$L(p) = (1-a)*p^b + a*\left(1 - (1-p)^{\frac{1}{b}}\right),\tag{10}$$

$$0 \le a \le 1,$$

$$1 \le b$$

**Table 8. The income data of the other 20 countries are used to demonstrate the performance of Paul and Shankar [3]'s functional form and those of the other popular functional forms based on R$^2$.**

| Country | Year | Income share of the top 20% | Coefficient of determination (R$^2$) | | | | |
|---|---|---|---|---|---|---|---|
| | | | PS | KP | A | C | Pareto |
| Thailand | 2017 | 51.12 | 0.9805 | 0.9875 | 0.9978 | 0.9899 | 0.9888 |
| Malawi | 2017 | 51.67 | 0.9608 | 0.9724 | 0.9911 | 0.9762 | 0.9969 |
| Ecuador | 2015 | 51.90 | 0.9801 | 0.9866 | 0.9977 | 0.9891 | 0.9888 |
| Benin | 2015 | 52.13 | 0.9811 | 0.9858 | 0.9958 | 0.9880 | 0.9862 |
| Philippines | 2012 | 52.68 | 0.9788 | 0.9855 | 0.9980 | 0.9883 | 0.9896 |
| Sri Lanka | 2010 | 54.40 | 0.9743 | 0.9795 | 0.9953 | 0.9826 | 0.9909 |
| Nicaragua | 2014 | 54.50 | 0.9772 | 0.9817 | 0.9962 | 0.9847 | 0.9893 |
| Bangladesh | 2016 | 54.75 | 0.9682 | 0.9736 | 0.9926 | 0.9772 | 0.9928 |
| Costa Rica | 2016 | 54.90 | 0.9842 | 0.9878 | 0.9986 | 0.9902 | 0.9847 |
| Saint Lucia | 2016 | 55.39 | 0.9857 | 0.9883 | 0.9982 | 0.9906 | **0.9827** |
| Paraguay | 2016 | 55.41 | 0.9824 | 0.9855 | 0.9978 | 0.9882 | 0.9857 |
| Mexico | 2016 | 55.70 | 0.9738 | 0.9783 | 0.9958 | 0.9817 | 0.9912 |
| India | 2012 | 56.38 | 0.9810 | 0.9839 | 0.9979 | 0.9868 | 0.9866 |
| Eswatini | 2010 | 56.73 | 0.9813 | 0.9842 | 0.9984 | 0.9871 | 0.9865 |
| Colombia | 2015 | 56.80 | 0.9815 | 0.9838 | 0.9977 | 0.9866 | 0.9859 |
| Brazil | 2018 | 58.60 | 0.9784 | 0.9797 | 0.9969 | 0.9829 | 0.9876 |
| Egypt | 2013 | 58.78 | 0.9726 | 0.9742 | 0.9953 | 0.9779 | 0.9908 |
| Mozambique | 2015 | 59.53 | 0.9647 | 0.9664 | 0.9929 | 0.9708 | 0.9943 |
| Honduras | 2012 | 60.10 | 0.9728 | 0.9734 | 0.9957 | 0.9772 | 0.9908 |
| Namibia | 2016 | 63.74 | 0.9810 | **0.9790** | 0.9985 | 0.9822 | 0.9854 |

PS, Paul and Shankar [3]; KP, Kakwani and Podder [5]; A, Aggarwal [9]; C, Chotikapanich [16]; Pareto, Functional form implied by Pareto distribution.

The bold numbers (if any) indicate that Paul and Shankar [3]'s functional form is superior to the other existing widely used functional forms.

This specified functional form satisfies all necessary and sufficient conditions for the Lorenz curve which are: $L(0) = 0$, $L(1) = 1$, $L(p)$ is convex, $\frac{dL}{dp} \geq 0$, and $\frac{d^2L}{dp^2} \geq 0$. According to Sitthiyot and Holasut [2], the parameter $b$ represents the degree of inequality in income distribution as measured by the Gini index. The parameter $a$ is the weight that controls the curvature of the Lorenz curve such that the Gini index remains constant since, for a given value of parameter $b$, there are infinite values of parameter $a$ that could give an identical value of the Gini index. The parameter $a$ thus provides the information about countries' income shares in case their Lorenz curves intersect. In addition, the shape of the estimated Lorenz curve could be conveniently adjusted through the change in parameter $a$ while the value of the Gini index is kept constant. The values of parameters $a$ and $b$ estimated using Sitthiyot and Holasut [2]'s functional form for the Lorenz curve are reported in Tables A1 and A2 in S1 Appendix. The estimated Lorenz curves of 40 countries are then used to calculate the values of the estimated Gini index which in turn would be compared to those computed using single-parameter functional forms considered in Paul and Shankar [3]'s study. Tables 13 and 14 report our results.

The results on the estimated Gini index shown in Tables 13 and 14 indicate that the two-parameter functional form proposed by Sitthiyot and Holasut [2], by and large, outperforms single-parameter functional forms used in Paul and Shankar [3]'s study. As shown in Tables 13 and 14, there are *only* 12 cases out of 200 cases where the single-parameter functional forms give the values of the estimated Gini index that are closer to their actual observations than the two-parameter functional form proposed by Sitthiyot and Holasut [2], all of which are from countries whose upper part of the income distribution has a long-tailed property. These results

**Table 9. The income data of the other 20 countries are used to demonstrate the performance of Paul and Shankar [3]'s functional form and those of the other popular functional forms based on MSE.**

| Country | Year | Income share of the top 20% | Mean-squared error (MSE) | | | | |
|---|---|---|---|---|---|---|---|
| | | | PS | KP | A | C | Pareto |
| Thailand | 2017 | 51.12 | 0.0013 | 0.0011 | 0.0003 | 0.0009 | 0.0006 |
| Malawi | 2017 | 51.67 | 0.0024 | 0.0021 | 0.0009 | 0.0019 | 0.0001 |
| Ecuador | 2015 | 51.90 | 0.0013 | 0.0011 | 0.0003 | 0.0010 | 0.0006 |
| Benin | 2015 | 52.13 | 0.0016 | 0.0015 | 0.0005 | 0.0013 | 0.0005 |
| Philippines | 2012 | 52.68 | 0.0013 | 0.0011 | 0.0002 | 0.0010 | 0.0006 |
| Sri Lanka | 2010 | 54.40 | 0.0019 | 0.0018 | 0.0006 | 0.0017 | 0.0004 |
| Nicaragua | 2014 | 54.50 | 0.0017 | 0.0017 | 0.0005 | 0.0015 | 0.0005 |
| Bangladesh | 2016 | 54.75 | 0.0025 | 0.0024 | 0.0009 | 0.0022 | 0.0003 |
| Costa Rica | 2016 | 54.90 | 0.0011 | 0.0010 | 0.0002 | 0.0009 | 0.0009 |
| Saint Lucia | 2016 | 55.39 | 0.0011 | 0.0011 | 0.0002 | 0.0009 | 0.0010 |
| Paraguay | 2016 | 55.41 | 0.0013 | 0.0013 | 0.0003 | 0.0012 | 0.0008 |
| Mexico | 2016 | 55.70 | 0.0019 | 0.0018 | 0.0005 | 0.0016 | 0.0004 |
| India | 2012 | 56.38 | 0.00137 | **0.00138** | 0.0003 | 0.0012 | 0.0008 |
| Eswatini | 2010 | 56.73 | 0.00127 | **0.00129** | 0.0002 | 0.0011 | 0.0008 |
| Colombia | 2015 | 56.80 | 0.00145 | **0.00149** | 0.0003 | 0.0013 | 0.0008 |
| Brazil | 2018 | 58.60 | 0.0017 | **0.0018** | 0.0004 | 0.0017 | 0.0006 |
| Egypt | 2013 | 58.78 | 0.0021 | **0.0023** | 0.0006 | 0.0021 | 0.0004 |
| Mozambique | 2015 | 59.53 | 0.0026 | **0.0028** | 0.0008 | 0.0025 | 0.0003 |
| Honduras | 2012 | 60.10 | 0.0021 | **0.0023** | 0.0005 | 0.0021 | 0.0005 |
| Namibia | 2016 | 63.74 | 0.0015 | **0.0018** | 0.0002 | **0.0016** | 0.0010 |

PS, Paul and Shankar [3]; KP, Kakwani and Podder [5]; A, Aggarwal [9]; C, Chotikapanich [16]; Pareto, Functional form implied by Pareto distribution.

The bold numbers (if any) indicate that Paul and Shankar [3]'s functional form is superior to the other existing widely used functional forms.

confirm that the two-parameter functional form, in general, performs better than one-parameter functional forms on the criterion of the estimated Gini index.

## Conclusions

Paul and Shankar [3] propose an alternative single-parameter functional form for the Lorenz curve and use the Australian income data between 2001 and 2010 that have the Gini index between 0.4442 and 0.4633 to show that their functional form is superior to the other existing widely used functional forms. Given that previous studies pointed out that an excellent performance of any parametric functional form for the Lorenz curve that is based on a single country case study and a limited range of distribution must be treated with extreme caution [32–34], this study demonstrates that when the observed Gini index is lower than the critical threshold which is found to be 0.4180 as for the cases of 20 OECD countries, the functional form proposed by Paul and Shankar [3] not only has a serious limitation in that it fails to fit the actual observations well but also is outperformed by the other existing widely used functional forms. This study also shows that when the upper part of income distribution has a long-tailed property as for the cases of the other 20 different countries, the functional form proposed by Paul and Shankar [3] is, by and large, outperformed by the other existing widely used functional forms considered in their study.

This study would like to note that, for future evaluation of any single parametric functional form for the Lorenz curve, the goodness-of-fit statistics should be given the priority since the shape of income distribution is more important than the estimated Gini index. As noted by

**Table 10. The income data of the other 20 countries are used to demonstrate the performance of Paul and Shankar [3]'s functional form and those of the other popular functional forms based on IIM.**

| Country | Year | Income share of the top 20% | Information inaccuracy measure (IIM) | | | | |
|---------|------|------------------------------|------|------|------|------|--------|
| | | | **PS** | **KP** | **A** | **C** | **Pareto** |
| Thailand | 2017 | 51.12 | 0.0494 | 0.0280 | 0.0085 | 0.0235 | 0.0237 |
| Malawi | 2017 | 51.67 | 0.0930 | 0.0577 | 0.0226 | 0.0505 | 0.0064 |
| Ecuador | 2015 | 51.90 | 0.0475 | 0.0278 | 0.0068 | 0.0230 | 0.0225 |
| Benin | 2015 | 52.13 | 0.0464 | 0.0377 | 0.0210 | 0.0346 | 0.0338 |
| Philippines | 2012 | 52.68 | 0.0556 | 0.0312 | 0.0064 | 0.0254 | 0.0223 |
| Sri Lanka | 2010 | 54.40 | 0.0631 | 0.0466 | 0.0162 | 0.0406 | 0.0202 |
| Nicaragua | 2014 | 54.50 | 0.0545 | 0.0411 | 0.0139 | 0.0357 | 0.0237 |
| Bangladesh | 2016 | 54.75 | 0.0750 | 0.0617 | 0.0266 | 0.0554 | 0.0190 |
| Costa Rica | 2016 | 54.90 | 0.0416 | 0.0291 | 0.0079 | 0.0241 | 0.0360 |
| Saint Lucia | 2016 | 55.39 | 0.0352 | 0.0284 | 0.0107 | 0.0244 | **0.0418** |
| Paraguay | 2016 | 55.41 | 0.0448 | 0.0340 | 0.0101 | 0.0288 | 0.0325 |
| Mexico | 2016 | 55.70 | 0.0648 | 0.0482 | 0.0134 | 0.0411 | 0.0195 |
| India | 2012 | 56.38 | 0.0475 | 0.0368 | 0.0092 | 0.0309 | 0.0312 |
| Eswatini | 2010 | 56.73 | 0.0491 | 0.0356 | 0.0061 | 0.0290 | 0.0304 |
| Colombia | 2015 | 56.80 | 0.0458 | 0.0372 | 0.0097 | 0.0314 | 0.0317 |
| Brazil | 2018 | 58.60 | 0.0551 | 0.0486 | 0.0141 | 0.0419 | 0.0309 |
| Egypt | 2013 | 58.78 | 0.0675 | 0.0593 | 0.0165 | 0.0514 | 0.0220 |
| Mozambique | 2015 | 59.53 | 0.0871 | 0.0747 | 0.0195 | 0.0645 | 0.0130 |
| Honduras | 2012 | 60.10 | 0.0697 | 0.0623 | 0.0153 | 0.0535 | 0.0231 |
| Namibia | 2016 | 63.74 | 0.0522 | **0.0524** | 0.0063 | 0.0432 | 0.0365 |

PS, Paul and Shankar [3]; KP, Kakwani and Podder [5]; A, Aggarwal [9]; C, Chotikapanich [16]; Pareto, Functional form implied by Pareto distribution.

The bold numbers (if any) indicate that Paul and Shankar [3]'s functional form is superior to the other existing widely used functional forms.

Dagum [35], whenever a functional form contains a single inequality parameter, the estimated Gini index would be a monotonic function of it, and hence, it would not be able to detect the intersecting Lorenz curves. To evaluate the performance on the criterion of the estimated Gini index, a functional form that contains more than one parameter is required since the curvature of the estimated Lorenz curve has to be adjustable so that it would fit the actual observations as much as possible while keeping the value of the estimated Gini index the same. To illustrate this point, this study employs the two-parameter functional form developed by Sitthiyot and Holasut [2] to estimate the Lorenz curve and compute the value of the estimated Gini index using the grouped data on income shares of 40 countries. The overall results indicate that the two-parameter functional form, by and large, performs better than the single-parameter functional forms considered in Paul and Shankar [3]'s study on the criterion of the value of the estimated Gini index.

Last but not least, before using any parametric functional form to estimate the Lorenz curve and calculate the Gini index, policymakers should be aware that the goodness-of-fit statistical measures, the shape of the estimated Lorenz curve, and the estimated Gini index should be taken into consideration altogether. The key lesson learned from Paul and Shankar [3]'s study is that their results should be treated with great caution since all of them rely on a single country case study and a narrow range of distribution [32–34]. As noted by Dagum [35], a good functional form for estimating the Lorenz curve must describe the shape of income distributions of different countries, regions, socioeconomic groups, and in different time periods through the changes in parameter values. Given that existing studies on the relationship

**Table 11. The income data of the other 20 countries are used to demonstrate the performance of Paul and Shankar [3]'s functional form and those of the other popular functional forms based on the estimated Gini index.**

| Country | Year | Income share of the top 20% | Observed Gini index | Estimated Gini index | | | | |
|---|---|---|---|---|---|---|---|---|
| | | | | PS | KP | A | C | Pareto |
| Thailand | 2017 | 51.12 | 0.4528 | 0.4669 | 0.4507 | 0.4469 | 0.4496 | 0.4390 |
| Malawi | 2017 | 51.67 | 0.4470 | 0.4600 | 0.4391 | 0.4404 | 0.4385 | 0.4345 |
| Ecuador | 2015 | 51.90 | 0.4620 | 0.4754 | 0.4594 | 0.4554 | 0.4582 | **0.4475** |
| Benin | 2015 | 52.13 | 0.4780 | 0.4842 | **0.4700** | **0.4656** | **0.4687** | **0.4585** |
| Philippines | 2012 | 52.68 | 0.4649 | 0.4812 | 0.4649 | 0.4607 | 0.4636 | 0.4526 |
| Sri Lanka | 2010 | 54.40 | 0.4900 | 0.5020 | 0.4865 | 0.4828 | 0.4850 | **0.4758** |
| Nicaragua | 2014 | 54.50 | 0.4950 | 0.5045 | 0.4898 | **0.4853** | 0.4883 | **0.4781** |
| Bangladesh | 2016 | 54.75 | 0.4980 | 0.5045 | **0.4891** | **0.4869** | **0.4877** | **0.4805** |
| Costa Rica | 2016 | 54.90 | 0.5000 | 0.5112 | 0.4982 | 0.4910 | 0.4965 | **0.4828** |
| Saint Lucia | 2016 | 55.39 | 0.5123 | 0.5191 | 0.5074 | **0.4994** | 0.5056 | **0.4915** |
| Paraguay | 2016 | 55.41 | 0.5100 | 0.5185 | 0.5058 | **0.4988** | 0.5040 | **0.4911** |
| Mexico | 2016 | 55.70 | 0.5040 | 0.5151 | 0.5001 | 0.4955 | 0.4984 | **0.4882** |
| India | 2012 | 56.38 | 0.5150 | 0.5262 | 0.5135 | 0.5063 | 0.5116 | **0.4986** |
| Eswatini | 2010 | 56.73 | 0.5145 | 0.5294 | 0.5167 | 0.5090 | 0.5147 | 0.5011 |
| Colombia | 2015 | 56.80 | 0.5240 | 0.5342 | 0.5222 | 0.5145 | 0.5202 | **0.5071** |
| Brazil | 2018 | 58.60 | 0.5240 | 0.5495 | 0.5380 | 0.5301 | 0.5358 | 0.5231 |
| Egypt | 2013 | 58.78 | 0.5400 | 0.5505 | 0.5378 | 0.5313 | 0.5356 | **0.5245** |
| Mozambique | 2015 | 59.53 | 0.5400 | 0.5545 | 0.5405 | 0.5352 | 0.5382 | 0.5283 |
| Honduras | 2012 | 60.10 | 0.5520 | 0.5633 | 0.5515 | 0.5437 | 0.5490 | **0.5369** |
| Namibia | 2016 | 63.74 | 0.5907 | 0.6052 | 0.5983 | 0.5846 | 0.5953 | 0.5791 |

PS, Paul and Shankar [3]; KP, Kakwani and Podder [5]; A, Aggarwal [9]; C, Chotikapanich [16]; Pareto, Functional form implied by Pareto distribution.

The bold numbers (if any) indicate that Paul and Shankar [3]'s functional form is superior to the other existing widely used functional forms.

between inequality measures and financial and/or socioeconomic variables rely on the accuracy of inequality measures as discussed in Introduction and many more to come in the future, if the choice of parametric functional form for the Lorenz curve is not a valid candidate for representing the income distribution, the estimates on the income shares and inequality measures could be severely affected by misspecification bias [32]. Thus, before applying any functional form for estimating the Lorenz curve, policymakers should carefully check whether or not it satisfies the aforementioned criteria suggested by Dagum [35]. This is because using a functional form that does not fit actual observations could adversely affect inequality measures and income distribution policies.

**Table 12. The income data of Thailand are used to demonstrate the performance of Paul and Shankar [3]'s functional form and those of the other popular functional forms all of which share the same value of the estimated Gini index which is equal to 0.4528.**

| | PS | KP | A | C | Pareto |
|---|---|---|---|---|---|
| **Estimated parameter** | $\gamma = 0.2273$ | $\delta = 3.5401$ | $\theta = 0.3482$ | $k = 3.1328$ | $\alpha = 2.6547$ |
| **R²** | 0.9797 | 0.9875 | 0.9977 | 0.9899 | 0.9882 |
| **MSE** | 0.0014 | 0.0010 | 0.0003 | 0.0009 | 0.0008 |
| **IIM** | 0.0498 | 0.0281 | 0.0083 | 0.0236 | 0.0260 |
| **Estimated Gini index** | 0.4528 | 0.4528 | 0.4528 | 0.4528 | 0.4528 |
| **Observed Gini index** | 0.4528 | | | | |

PS, Paul and Shankar [3]; KP, Kakwani and Podder [5]; A, Aggarwal [9]; C, Chotikapanich [16]; Pareto, Functional form implied by Pareto distribution.

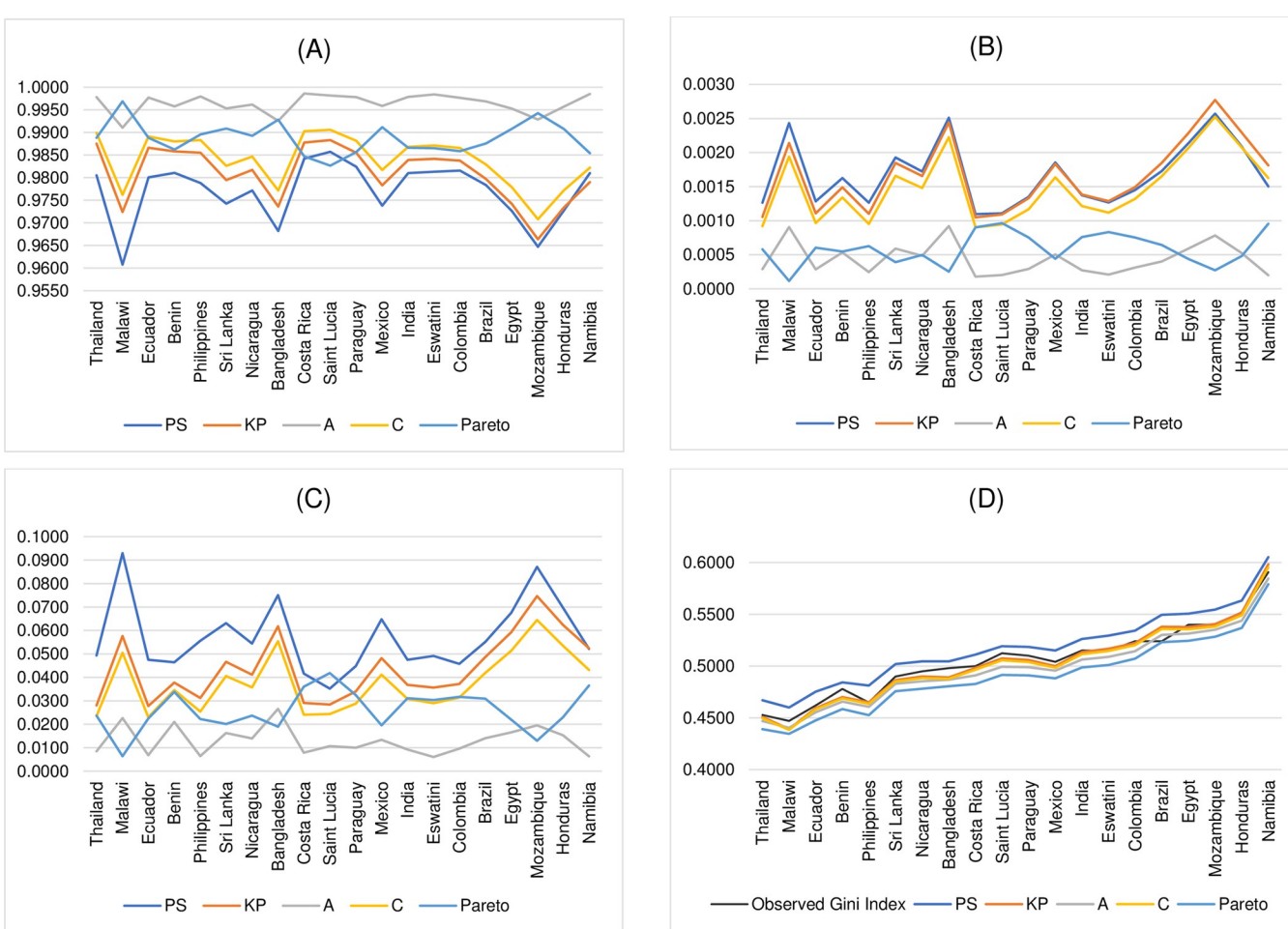

**Fig 3. The performance comparison between Paul and Shankar [3]'s functional form and the other existing widely used functional forms, namely, Kakwani and Podder [5], Aggarwal [9], Chotikapanich [16], and functional form implied by Pareto distribution when the income data has a long-tailed property.** (A) $R^2$. (B) MSE. (C) IIM. (D) Estimated Gini index.

**Table 13. The income data of 20 OECD countries are used to demonstrate the performance of single-parameter functional forms and that of a two-parameter functional form proposed by Sitthiyot and Holasut [2] on the criterion of estimated Gini index.**

| Country | Year | Observed Gini index | Estimated Gini index | | | | | |
|---|---|---|---|---|---|---|---|---|
| | | | PS | KP | A | C | Pareto | SH |
| Slovakia | 2017 | 0.2320 | 0.4180 | 0.2242 | 0.2227 | 0.2239 | 0.2171 | 0.2300 |
| Slovenia | 2017 | 0.2370 | 0.4180 | 0.2315 | 0.2305 | 0.2313 | 0.2250 | 0.2372 |
| Czech Republic | 2017 | 0.2450 | 0.4180 | 0.2386 | 0.2389 | 0.2387 | 0.2351 | 0.2445 |
| Finland | 2017 | 0.2530 | 0.4180 | 0.2470 | 0.2470 | 0.2470 | 0.2426 | 0.2530 |
| Belgium | 2017 | 0.2600 | 0.4180 | 0.2540 | 0.2526 | 0.2538 | 0.2463 | 0.2602 |
| Norway | 2017 | 0.2610 | 0.4180 | 0.2520 | 0.2514 | 0.2518 | 0.2467 | 0.2583 |
| Netherlands | 2017 | 0.2710 | 0.4180 | 0.2628 | 0.2622 | 0.2627 | 0.2568 | 0.2690 |
| Denmark | 2017 | 0.2760 | 0.4180 | 0.2646 | 0.2644 | 0.2646 | 0.2599 | 0.2711 |
| Austria | 2017 | 0.2790 | 0.4180 | 0.2711 | 0.2701 | 0.2709 | 0.2644 | 0.2776 |
| Sweden | 2017 | 0.2800 | 0.4180 | 0.2709 | 0.2697 | 0.2706 | 0.2639 | 0.2773 |
| Hungary | 2017 | 0.2810 | 0.4180 | 0.2739 | 0.2728 | 0.2737 | 0.2669 | 0.2802 |

*(Continued)*

**Table 13.** (Continued)

| Country | Year | Observed Gini index | Estimated Gini index | | | | | |
|---------|------|---------------------|------|------|------|------|--------|------|
| | | | PS | KP | A | C | Pareto | SH |
| Germany | 2017 | 0.2910 | 0.4180 | 0.2833 | 0.2824 | 0.2831 | 0.2764 | 0.2898 |
| Poland | 2017 | 0.2920 | 0.4180 | 0.2843 | 0.2835 | 0.2842 | 0.2777 | 0.2907 |
| France | 2017 | 0.2930 | 0.4180 | 0.2834 | 0.2837 | 0.2834 | 0.2794 | 0.2899 |
| Switzerland | 2017 | 0.3010 | 0.4180 | 0.2927 | 0.2924 | 0.2926 | 0.2871 | 0.2992 |
| Ireland | 2017 | 0.3060 | 0.4180 | 0.3002 | 0.2995 | 0.3001 | 0.2933 | 0.3065 |
| Canada | 2017 | 0.3090 | 0.4180 | 0.3027 | 0.3004 | 0.3022 | 0.2929 | 0.3093 |
| Luxembourg | 2017 | 0.3090 | 0.4180 | 0.3037 | 0.3019 | 0.3034 | 0.2948 | 0.3102 |
| Estonia | 2017 | 0.3160 | 0.4180 | 0.3120 | 0.3087 | 0.3114 | 0.2999 | 0.3187 |
| Italy | 2017 | 0.3270 | 0.4180 | 0.3183 | 0.3156 | 0.3178 | 0.3079 | 0.3251 |

PS, Paul and Shankar [3]; KP, Kakwani and Podder [5]; A, Aggarwal [9]; C, Chotikapanich [16]; Pareto, Functional form implied by Pareto distribution; SH, Sitthiyot and Holasut [2].

The bold numbers (if any) indicate that the single-parameter functional forms are superior to the two-parameter function form.

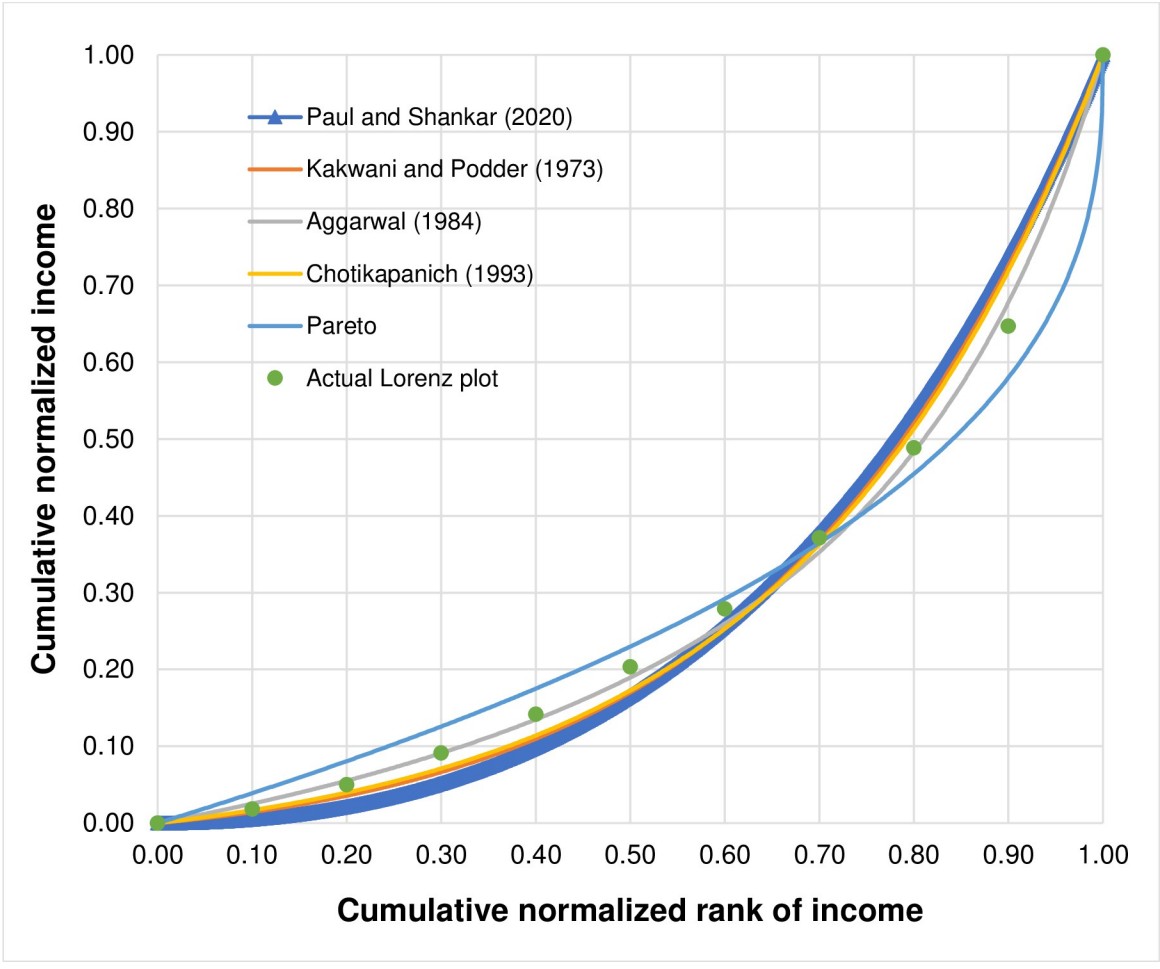

**Fig 4. The actual Lorenz plot and the estimated Lorenz curves using different functional forms all of which share the same value of the estimated Gini index which is equal to 0.4528.**

**Table 14. The income data of the other 20 countries are used to demonstrate the performance of single-parameter functional forms and that of a two-parameter functional form proposed by Sitthiyot and Holasut [2] on the criterion of estimated Gini index.**

| Country | Year | Observed Gini index | Estimated Gini index | | | | | |
|---|---|---|---|---|---|---|---|---|
| | | | PS | KP | A | C | Pareto | SH |
| Thailand | 2017 | 0.4528 | 0.4669 | **0.4507** | 0.4469 | 0.4496 | 0.4390 | 0.4548 |
| Malawi | 2017 | 0.4470 | 0.4600 | 0.4391 | 0.4404 | 0.4385 | 0.4345 | 0.4425 |
| Ecuador | 2015 | 0.4620 | 0.4754 | 0.4594 | 0.4554 | 0.4582 | 0.4475 | 0.4632 |
| Benin | 2015 | 0.4780 | 0.4842 | 0.4700 | 0.4656 | 0.4687 | 0.4585 | 0.4742 |
| Philippines | 2012 | 0.4649 | 0.4812 | **0.4649** | 0.4607 | **0.4636** | 0.4526 | 0.4680 |
| Sri Lanka | 2010 | 0.4900 | 0.5020 | 0.4865 | 0.4828 | 0.4850 | 0.4758 | 0.4889 |
| Nicaragua | 2014 | 0.4950 | 0.5045 | 0.4898 | 0.4853 | 0.4883 | 0.4781 | 0.4924 |
| Bangladesh | 2016 | 0.4980 | **0.5045** | 0.4891 | 0.4869 | 0.4877 | 0.4805 | 0.4913 |
| Costa Rica | 2016 | 0.5000 | 0.5112 | 0.4982 | 0.4910 | 0.4965 | 0.4828 | 0.5008 |
| Saint Lucia | 2016 | 0.5123 | 0.5191 | 0.5074 | 0.4994 | 0.5056 | 0.4915 | 0.5100 |
| Paraguay | 2016 | 0.5100 | 0.5185 | 0.5058 | 0.4988 | 0.5040 | 0.4911 | 0.5080 |
| Mexico | 2016 | 0.5040 | 0.5151 | 0.5001 | 0.4955 | 0.4984 | 0.4882 | 0.5014 |
| India | 2012 | 0.5150 | 0.5262 | 0.5135 | 0.5063 | 0.5116 | 0.4986 | 0.5149 |
| Eswatini | 2010 | 0.5145 | 0.5294 | **0.5167** | 0.5090 | **0.5147** | 0.5011 | 0.5178 |
| Colombia | 2015 | 0.5240 | 0.5342 | 0.5222 | 0.5145 | 0.5202 | 0.5071 | 0.5235 |
| Brazil | 2018 | 0.5240 | 0.5495 | 0.5380 | **0.5301** | **0.5358** | **0.5231** | 0.5379 |
| Egypt | 2013 | 0.5400 | 0.5505 | **0.5378** | 0.5313 | 0.5356 | 0.5245 | 0.5370 |
| Mozambique | 2015 | 0.5400 | 0.5545 | **0.5405** | 0.5352 | 0.5382 | 0.5283 | 0.5383 |
| Honduras | 2012 | 0.5520 | 0.5633 | **0.5515** | 0.5437 | 0.5490 | 0.5369 | 0.5495 |
| Namibia | 2016 | 0.5907 | 0.6052 | 0.5983 | 0.5846 | 0.5953 | 0.5791 | 0.5943 |

PS, Paul and Shankar [3]; KP, Kakwani and Podder [5]; A, Aggarwal [9]; C, Chotikapanich [16]; Pareto, Functional form implied by Pareto distribution; SH, Sitthiyot and Holasut [2].

The bold numbers (if any) indicate that the single-parameter functional forms are superior to the two-parameter function form.

## Supporting information

**S1 Appendix.**
(DOCX)

## Acknowledgments

The authors are grateful to Dr. Suradit Holasut for guidance and comments.

## Author Contributions

**Conceptualization:** Thitithep Sitthiyot.

**Formal analysis:** Thitithep Sitthiyot.

**Methodology:** Thitithep Sitthiyot, Kanyarat Holasut.

**Validation:** Kanyarat Holasut.

**Writing – original draft:** Thitithep Sitthiyot.

**Writing – review & editing:** Thitithep Sitthiyot, Kanyarat Holasut.

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
