## [Decision Letter · Decision Letter 0]

18 Nov 2022

PONE-D-22-23738An investigation of the performance of parametric functional forms for the Lorenz curvePLOS ONE

Dear Dr. Authors

Thank you for submitting your manuscript to PLOS ONE. After careful consideration, we feel that it has merit but does not fully meet PLOS ONE’s publication criteria as it currently stands. Therefore, we invite you to submit a revised version of the manuscript that addresses the points raised during the review process.

 After considering the reviewer's comments, the article will be suitable for publication, but major revision is needed

We look forward to receiving your revised manuscript.

Kind regards,

Yuriy Bilan

Academic Editor

PLOS ONE

Journal Requirements:

I have read the journal's policy and the authors of this manuscript have the following competing interests: 

The findings reported in this manuscript contradict the results previously published in Paul and Shankar (2020).

Additional Editor Comments:

After considering the reviewer's comments, the article will be suitable for publication

but major revision is needed

Note from Staff Editor Hanna Landenmark (hlandenmark@plos.org): 1) Please note that there is no requirement to cite the specific articles suggested by the reviewers, but please take into account their request for further context for this study. 2) Please note that it is policy of PLOS ONE to invite a signed review from the authors of the disputed work; this review will be considered by the Academic Editor in the decision process, but the editor will be made aware of the competing interests (http://journals.plos.org/plosone/s/submission-guidelines#loc-manuscripts-disputing-published-work). This invitation will take place following the first round of independent peer review.

Reviewers' comments:

Reviewer's Responses to Questions

**Comments to the Author**

1. Is the manuscript technically sound, and do the data support the conclusions?

Reviewer #1: Partly

Reviewer #2: Yes

2. Has the statistical analysis been performed appropriately and rigorously? 

Reviewer #1: Yes

Reviewer #2: Yes

3. Have the authors made all data underlying the findings in their manuscript fully available?

Reviewer #1: Yes

Reviewer #2: Yes

4. Is the manuscript presented in an intelligible fashion and written in standard English?

Reviewer #1: Yes

Reviewer #2: Yes

5. Review Comments to the Author

Reviewer #1: There is actually no "Introduction" section. Currently, a review of the literature is carried out here, which should be updated according to the latest scientific results.

It is worth adding the relevance and purpose of the research.

Other comments are given in the article.

Reviewer #2: The article is interesting in terms of topics, but not entirely correctly implemented in terms of structure. It is noticeable at the beginning that there is no review of world research in this area, i.e. the so-called Review of the literature. Throughout the article, the authors refer to only 15 items !!! This needs to be completely rewritten and corrected. Since you analyze, for example, the Gini index, it is worth referring to such items as:

1. Statistical methods of the bankruptcy prediction in the logistics sector in Poland and Slovakia, Transformations in Business and Economics Vol. 15 (No 1 (37)): 93-114;

2. PREDICTING BANKRUPTCY OF COMPANIES FROM THE LOGISTICS SECTOR OPERATING IN THE PODKARPACIE REGION; doi: 10.7862 / rz.2013.mmr.33

3. Evaluating the impact of GINI index and information gain on classification using decision tree classifier algorithm; doi: 10.14569 / ijacsa.2020.0110277

4. The gini index: A proposal for revision; doi: 10.24309 / recta.2020.21.1.01

or others.

Authors must also emphasize the so-called added value of research. What's new in these findings?

6. PLOS authors have the option to publish the peer review history of their article (what does this mean?). If published, this will include your full peer review and any attached files.

Reviewer #1: No

Reviewer #2: No

---

## [Author Response · Author response to Decision Letter 0]

19 Dec 2022

Dear Reviewers #1 and #2,

We sincerely thank both Reviewers for pointing out several important concerns and providing useful suggestions. We have made a major revision by addressing all comments and suggestions made by both Reviewers and tried our best to incorporate them into our revised manuscript. We hope that our revised manuscript is clearer in all aspects that the Reviewers have concerned and/or suggested. Let us respond to the Reviewers’ concerns and suggestions as follows.

Reviewer #1: 

- There is actually no "Introduction" section. Currently, a review of the literature is carried out here, which should be updated according to the latest scientific results. 

We sincerely thank Reviewer #1 for comment on this very important point. We would like to inform Reviewer #1 that we have made a substantial revision in Introduction by providing definition of the Lorenz curve in paragraph 1 and discussing how the Gini index can be calculated from the Lorenz curve in paragraph 2 in our revised manuscript. We also explain in Introduction, paragraph 3 in our revised manuscript how the Lorenz curve can be estimated as well as provide the list of up-to-date studies that propose different parametric functional forms for estimating the Lorenz curve.

- It is worth adding the relevance and purpose of the research.

When justifying its relevance, it is worth pointing out which economic and social processes are affected by income inequality, why is it important to study it?

For example: Impact of Income Distribution on Social and Economic Well-Being of the State https://doi.org/10.3390/su12010429

In response to Reviewer #1’s suggestion to add the relevance and purpose of our study, we state the purpose of our study in Introduction, paragraph 4 in our revised manuscript which is to investigate an alternative single-parameter functional form for the Lorenz curve proposed by Paul and Shankar (2020) who show that their functional form outperforms the other 4 popular parametric functional forms for the Lorenz curve, namely, Kakwani and Podder (1973), Aggarwal (1984), Chotikapanich (1993), and a functional form implied by Pareto distribution. 

In addition, we state the relevance and contribution of our study in Introduction, paragraph 5 in our revised manuscript that it is relevant and worthwhile to conduct an investigation to find out if we use grouped income data of other countries, the performance of Paul and Shankar (2020)’s functional form is still superior to the other existing widely used functional forms considered in their study. The findings from this investigation should also contribute as a check-and-balance not only for economics but also for other scientific disciplines that use the Lorenz curve to analyze size distributions of non-negative quantities and inequalities.

Since the main focus of our study is to investigate the performance of parametric functional forms for estimating the Lorenz curve and to examine whether or not those parametric functional forms have an explicit mathematical solution for the Gini index, we provide an additional justification for our study as noted by Reviewer #1 and also by PLOS ONE Staff Editor Dr. Hanna Landenmark by noting in Introduction, paragraph 5 and in Conclusions, paragraph 3 in our revised manuscript that various studies on the relationship between inequality measures and financial and/or socioeconomic variables such as Kharlamova et al. (2018), Bilan et al. (2020), and Tung (2020) as suggested by Reviewer #1 and Pisular et al. (2013) as suggested by Reviewer #2, rely on the accuracy of inequality measures that could possibly be derived from a parametric functional form for the Lorenz curve. If the choice of parametric functional form is not a valid candidate for representing the income distribution, the estimates on the income shares and inequality measures might be severely affected by misspecification bias. This justification is reiterated in Conclusions, paragraph 3 in our revised manuscript.

- In the literature review, it is worth investigating the relationship between income inequality and other parameters (social, economic, etc.). 

For example: ANALYSIS OF THE RELATIONSHIP BETWEEN INCOME INEQUALITY AND SOCIAL VARIABLES: EVIDENCE FROM INDONESIA DOI: 10.14254/2071-789X.2021/14-1/7

THE RELATIONSHIP BETWEEN INCOME GROWTH AND INEQUALITY: EVIDENCE FROM AN ASIAN EMERGING ECONOMY DOI: 10.14254/2071-789X.2022/15-2/6

THE IMPACT OF TECHNOLOGICAL CHANGES ON INCOME INEQUALITY: THE EU STATES CASE STUDY DOI: 10.14254/2071-8330.2018/11-2/6

In response to the issue of investigating studies on the relationship between income inequality and other parameters as suggested by Reviewer #1, we went over those studies and find them very useful. However, given that the main focus of our study is to investigate the performance of parametric functional forms for estimating the Lorenz curve and to examine whether or not those parametric functional forms considered in Paul and Shankar (2020) study have a closed-form expression for the Gini index, we therefore include studies by Kharlamova et al. (2018), Bilan et al. (2020), Tung (2020) as suggested by Reviewer #1 and Pisular et al. (2013) as suggested by Reviewer #2 in Introduction, paragraph 5 in our revised manuscript in order to illustrate the importance of using the appropriate parametric functional form for estimating the Lorenz curve. We also mentioned the importance of this issue in Conclusions, paragraph 3 in our revised manuscript.

- Discussion should be separated into a separate section.

We follow Reviewer #1’s suggestion by deleting “discussion” in our revised manuscript.

- It is worth noting other publications in which the Gini index is considered. 

For example: CAN PUBLIC DEBT HARM SOCIAL DEVELOPMENT? EVIDENCE FROM THE ASIAN-PACIFIC REGION doi:10.14254/2071-8330.2020/13-2/4

We follow Reviewer #1’s suggestion by including study by Tung (2020) in Introduction, paragraph 5 and reiterating its relevance to our study in Conclusions, paragraph 3 in our revised manuscript.

- It is necessary to update according to the latest research, to issue according to the requirements.

We sincerely thank Reviewer #1 for pointing this out. We cite the up-to-date studies that propose different parametric functional forms for estimating the Lorenz curve in Introduction, paragraph 3 and include them in References in our revised manuscript.

Reviewer #2: 

- The article is interesting in terms of topics, but not entirely correctly implemented in terms of structure. It is noticeable at the beginning that there is no review of world research in this area, i.e. the so-called Review of the literature. Throughout the article, the authors refer to only 15 items !!! This needs to be completely rewritten and corrected. Since you analyze, for example, the Gini index, it is worth referring to such items as:

1. Statistical methods of the bankruptcy prediction in the logistics sector in Poland and Slovakia, Transformations in Business and Economics Vol. 15 (No 1 (37)): 93-114;

2. PREDICTING BANKRUPTCY OF COMPANIES FROM THE LOGISTICS SECTOR OPERATING IN THE PODKARPACIE REGION; doi: 10.7862 / rz.2013.mmr.33

3. Evaluating the impact of GINI index and information gain on classification using decision tree classifier algorithm; doi: 10.14569 / ijacsa.2020.0110277

4. The gini index: A proposal for revision; doi: 10.24309/recta.2020.21.1.01

or others.

We sincerely thanks Reviewer #2 for comment and suggestion regarding the review of world research in the area of parametric functional forms for estimating the Lorenz curve. In our revised manuscript, we have made a major revision by citing the up-to-date studies in Introduction, paragraph 3 and including those studies in References. We also rewrite Introduction by providing the definition of the Lorenz curve and its applications in paragraph 1. In addition, we explain how the Gini index can be derived from the Lorenz curve in Introduction, paragraph 2. In Introduction, paragraph 5 and Conclusions, paragraph 3, studies by Pisula et al. (2013) as suggested by Reviewer #2 and Kharlamova et al. (2018), Bilan et al. (2020), and Tung (2020) as suggested by Reviewer #1, all of which employ the Gini index in their analyses, are included as a part of our discussion on the importance of using an appropriate parametric functional form for estimating the Lorenz curve and calculating the Gini index.

- Authors must also emphasize the so-called added value of research. What's new in these findings?

In response to Reviewer #2’ s concern on the issue of the added value of our study, we state in Introduction, paragraph 5 in our revised manuscript that, given the superiority of Paul and Shankar (2020)’s functional form over the other existing widely used functional forms, it is therefore relevant and worthwhile to conduct an examination to find out if we use grouped income data of other countries, the performance of Paul and Shankar (2020)’s functional form is still superior to the other existing widely used functional forms considered in their study. The findings from our investigation should also contribute as a check-and-balance not only for economics but also for other scientific disciplines that use the Lorenz curve to analyze size distributions of non-negative quantities and inequalities. This is because using a functional form that does not fit the actual observations could adversely affect inequality measures and income distribution policies. 

In addition to new findings reported in Results, we also summarize our main findings in Introduction, paragraphs 6 and 7 and in Conclusions, paragraphs 1 and 2 in our revised manuscript as suggested by Reviewer #2.

---

## [Decision Letter · Decision Letter 1]

8 Jun 2023

An investigation of the performance of parametric functional forms for the Lorenz curve

PONE-D-22-23738R1

Dear Dr. Sitthiyot,

We’re pleased to inform you that your manuscript has been judged scientifically suitable for publication and will be formally accepted for publication once it meets all outstanding technical requirements.

Kind regards,

Dr Ayesha Afzal

Academic Editor

PLOS ONE

Additional Editor Comments (optional):

Reviewers' comments:

Reviewer's Responses to Questions

**Comments to the Author**

1. If the authors have adequately addressed your comments raised in a previous round of review and you feel that this manuscript is now acceptable for publication, you may indicate that here to bypass the “Comments to the Author” section, enter your conflict of interest statement in the “Confidential to Editor” section, and submit your "Accept" recommendation.

Reviewer #1: (No Response)

Reviewer #3: All comments have been addressed

2. Is the manuscript technically sound, and do the data support the conclusions?

Reviewer #1: Yes

Reviewer #3: Yes

3. Has the statistical analysis been performed appropriately and rigorously? 

Reviewer #1: Yes

Reviewer #3: Yes

4. Have the authors made all data underlying the findings in their manuscript fully available?

Reviewer #1: Yes

Reviewer #3: Yes

5. Is the manuscript presented in an intelligible fashion and written in standard English?

Reviewer #1: Yes

Reviewer #3: Yes

6. Review Comments to the Author

Reviewer #1: The authors of this scientific article made appropriate corrections according to the comments.

Reviewer #3: This is an interesting study; however, it is limited in terms of the authors discussing the practical implications of their findings. Therefore, they should address this concern before I can recommend the article for publication.

7. PLOS authors have the option to publish the peer review history of their article (what does this mean?). If published, this will include your full peer review and any attached files.

Reviewer #1: No

Reviewer #3: No

---

## [Editor Report · Acceptance letter]

14 Jun 2023

PONE-D-22-23738R1 

An investigation of the performance of parametric functional forms for the Lorenz curve 

Dear Dr. Sitthiyot:

I'm pleased to inform you that your manuscript has been deemed suitable for publication in PLOS ONE. Congratulations! Your manuscript is now with our production department. 

Kind regards, 

on behalf of

Dr. Ayesha Afzal 

Academic Editor

PLOS ONE